



# Distinct ionospheric long-term trends in Antarctica due to the Weddell Sea Anomaly

Marayén Canales[1], Trinidad Duran[2], Manuel Bravo[3], Andriy Zalizovski[4], Alberto Foppiano[3]

[1] Departamento de Geofísica, Universidad de Concepción, Concepción, 4070386, Chile
[2] Departamento de Física, Universidad Nacional del Sur, Bahía Blanca, 8000, Argentina
[3] Centro de Instrumentación Científica, Universidad Adventista de Chile, Chillán, 3780000, Chile
[4] Institute of Radio Astronomy, National Academy of Sciences of Ukraine, Kharkiv, 61002, Ukraine

*Correspondence to*: Marayén Canales (mcanales2019@udec.cl)

**Abstract.** The Weddell Sea Anomaly (WSA), a summer ionospheric anomaly over the eastern Antarctic Peninsula, was first observed in 1958 and is characterized by a nighttime peak in electron concentration, unlike the typical daytime peak. There are some works that examine long-term trends at ionospheric stations in the WSA region but they do not do a seasonal-diurnal analysis that is vital for differentiating the periods of the anomaly. This study investigates the seasonal-diurnal variation of the long-term trend in the F2 layer critical frequency (foF2) at ionospheric stations located within the WSA region: Vernadsky (Argentine Island; 65.1°S, 64.2°W) and Port Stanley (51.6°S, 57.9°W), both with long-term foF2 data. Data from Vernadsky (1960-2023) and Port Stanley (1960-2019) were analyzed alongside data from Syowa (69.0°S; 39.6°E) and Mawson (67.6°S; 62.9°E), two stations outside the WSA influence. The analysis reveals distinct seasonal and diurnal trends. For Vernadsky, negative foF2 trends (-0.02 MHz/year) are observed during summer nights, coinciding with the WSA's presence. Port Stanley shows similar trends but with a secondary nighttime maximum. The WSA's influence on Vernadsky is more pronounced, with Port Stanley exhibiting a weaker, mid-latitude summer evening anomaly. In contrast, Syowa and Mawson show different trends, with Syowa without a clear trend pattern, and Mawson showing negative trends throughout the year. The study concludes that the WSA significantly affects Vernadsky and, to a lesser extent, Port Stanley. The findings highlight regional variations in ionospheric behavior and contribute to the ongoing discussion on global ionospheric trends, suggesting that local phenomena like the WSA can modulate these trends.



## 1 Introduction


The Weddell Sea Anomaly (WSA) is a summer abnormality in the ionosphere over the eastern Antarctic Peninsula,
characterized by maximum electron concentration occurring during nighttime hours instead of the typical daytime peak. The
anomaly was first observed by Bellchambers and Piggott (1958) at the Halley Bay ionosonde (75.5°S; 26.6°W) in Antarctica,
located along the coast of the Weddell Sea. More recently, Total Electron Concentration (TEC) determined from satellite
measurements has shown this anomaly over the geographical region of 55°S to 75°S latitude and 80°W to 30°W longitude
(Zakharenkova et al., 2017).
We detected two ionosondes located in the WSA region with foF2 (F2 layer critical frequency) data records extensive enough
to analyze long-term trends linked to the anthropogenic activity: Argentine Island, also called Vernadsky (65.1°S, 64.2°W)
and Port Stanley (51.6°S, 57.9°W) located on the northern edge of the WSA. These trends have been of interest since a
pioneering study in 1989 suggesting that the long-term increase of greenhouse gases concentration due to anthropogenic
activity, particularly carbon dioxide, would produce a global cooling in the upper atmosphere in conjunction with the global
warming in the troposphere (Roble and Dickinson, 1989; Rishbeth, 1990). Since then, long-term changes in the upper
atmosphere, and particularly in the ionosphere, have become a significant topic in global change research with many results
already published as can be appreciated in the review works by Lastovicka and different co-authors (Lastovicka et al., 2012,
2014; Lastovicka, 2017, 2021a). Among these studies, we highlight those including the analysis of ionospheric stations located
within the WSA region.
The first study reporting trends at Port Stanley is that by Upadhyay and Mahajan (1998). Considering the period 1957-1990
and noon time hours they obtained an hmF2 (peak height of the F2 layer) trend of -0.33 km/year and a foF2 trend of -0.004
MHz/year. A year later, Jarvis et al. (1998) analyzed hmF2 at Argentine Island and Port Stanley along the period 1957-1995.
The trends obtained in this work, which are mostly negative, vary with month and time of day at both sites. They interpreted
these results either as a constant decrease in altitude combined with a decreasing thermospheric wind effect or as a constant
decrease in altitude which is altitude-dependent. Both interpretations left inconsistencies when the results from the two sites
are compared at that time, but the estimated long-term hmF2 decrease along the period considered was of a similar order of
magnitude to that which has been predicted to result in the thermosphere from anthropogenic greenhouse gas increase. There
is no mention of the WSA, but this is expected since the anomaly is seen in foF2 daily variation, and not in hmF2. It is worth
mentioning that Alfonsi et al. (2001) tried to analyze Halley Bay trend, but after detecting errors in foF2 data series in the
period 1957-1990 it was discarded from the study.
Some of these stations have been included in later studies, such as Bremer et al. (2012), which conducted a global analysis
considering the Damboldt and Suessman database (Damboldt and Suessman, 2012) which covers up to ~2009, but again no
distinction is made about any anomalies and no markedly regional dependencies in trend values are found.
In the present study, the diurnal and seasonal variation of foF2 long-term trend is analyzed for stations within the Weddell Sea
anomaly region to contribute to the still controversial ionospheric trend topic and the detection and attribution of their forcings.



**2 Data**

The extensive dataset of monthly median foF2 from the Vernadsky ionospheric station, covering the period from 1996 to 2023. Data spanning from 1960 to 1995 were sourced from the database made available by Damboldt and Suessmann (2012) in the Australian Space Weather Forecasting Centre (www.sws.bom.gov.au). The Vernadsky Academician's station is a Ukrainian research station in Antarctica, located at Marina Point on Galindez Island in the Argentine Island group of the Wilhelm Archipelago (see Fig. 1). It was previously the Faraday Base (or F Base) of the United Kingdom, which transferred it to Ukraine in 1996.

The Port Stanley dataset covering 1960 to October 2006 was obtained from the database made available by Damboldt and Suessmann (2012), but it was supplemented with digisonde data from the Digital Ionogram Data Base (DIDBase, https://giro.uml.edu), extending it until February 2019.

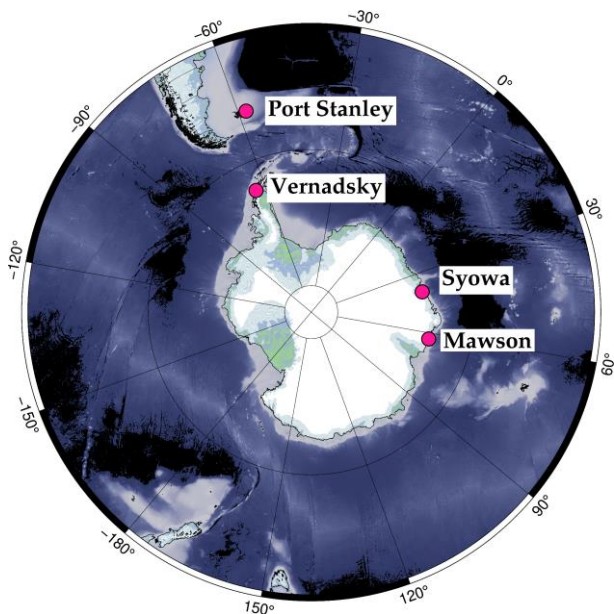

**Figure 1: Geographic locations of the ionospheric stations used in this work.**

To investigate the potential impact of the Weddell Sea anomaly on Vernadsky and Port Stanley ionospheric stations, the methodology will be applied to stations located outside the anomaly's influence zone. These stations are Syowa (69.0°S; 39.6°E) and Mawson (67.6°S; 62.9°E). Both datasets were also obtained from the database of Damboldt and Suessmann (2012). The geophysical information of each station is presented in Table 1. The data used begins in 1960, as it was decided to homogenize the study period to ensure a more consistent and precise comparison, avoiding the great solar maximum of 1958.



**Table 1: Geophysical information of ionospheric stations used in this work, according to the British Geological Survey (https://geomag.bgs.ac.uk/).**

| Ionospheric station | Geographical coordinates | Geomagnetic coordinates | Period |
|---|---|---|---|
| Vernadsky | 65.1°S; 64.2°W | 51.4°S; 9.2°E | Jan 1960 - Dec 2023 |
| Port Stanley | 51.6°S; 57.9°W | 40.0°S; 10.6°E | Jan 1960 - Oct 2006 + Nov 2006 - Feb 2019 |
| Syowa | 69.0°S; 39.6°E | 66.6°S; 73.8°E | Jan 1960 - Dec 2023 |
| Mawson | 67.6°S; 62.9°E | 70.6°S; 92.6°E | Jan 1960 - Dec 2023 |

Monthly median foF2 for each of the 24 daily hours were considered along the period January 1960-December 2023 of each station, except for Port Stanley which covers the period 1960-2019. The presence of the WSA becomes evident at Vernadsky when comparing its summer diurnal foF2 variations with Syowa and Mawson stations (see Fig. 2). However, Port Stanley is not completely affected by the WSA, but there is only a secondary maximum at night, or what is known as Mid-latitude Summer Evening Anomaly (MSEA) (Klimenko et al., 2015).

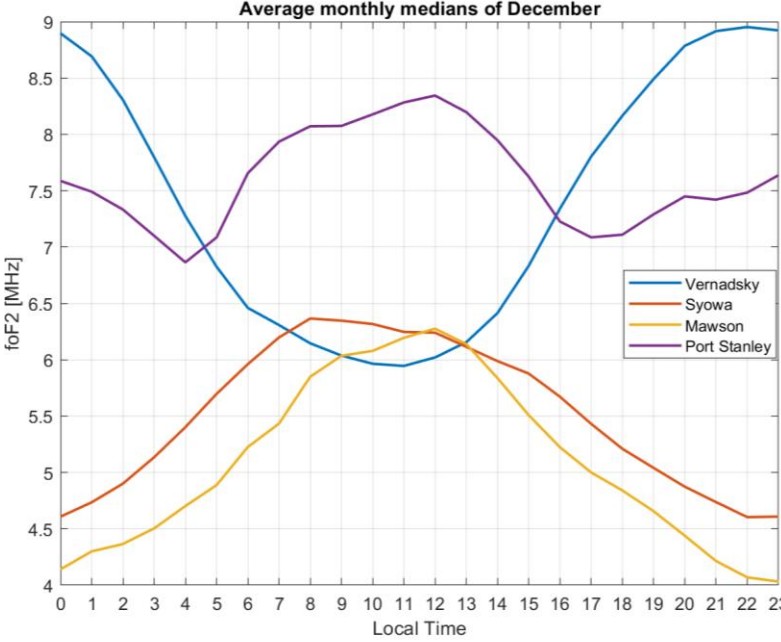

**Figure 2: Diurnal variation in December of the average monthly medians of foF2 for the period between 1960 and 2023, for Vernadsky, Syowa and Mawson stations and between 1960 and 2019 for Port Stanley station.**



It is important to note that the data for Port Stanley come from two different sources: most of the records up to 2006 were
obtained using an ionosonde, while from that date onwards they began to be collected using a digisonde. When comparing the
data for the same years, a satisfactory agreement was observed between both sources, which led to the decision to use them
together. It is even more relevant to assess the quality of data from the Mawson and Syowa ionospheric stations, especially
considering that in some years complete records are not available. Possible deficiencies and missing data from these stations
are presumed to be due to their proximity to the auroral oval, a highly dynamic region where geomagnetic conditions can
significantly interfere with ionospheric measurements.
Monthly means of MgII (core-to-wing ratio of Mg II line), as an EUV solar proxy was used to filter out solar activity effect
from foF2. It was chosen in accordance with the recommendations provided by Laštovička (2021a, 2021b) and de Haro Barbas
et al. (2021). The MgII index is available from the University of Bremen at http://www.iup.uni-
bremen.de/UVSAT/datasets/mgii (Viereck et al., 2004; Snow et al., 2014).
The geomagnetic activity index Ap was also considered as an additional parameter in the filtering process. Monthly values
were obtained from the Kyoto World Data Center for Geomagnetism at https://wdc.kugi.kyoto-u.ac.jp/index.html.

**3. Methodology**
Given that the foF2 trends we aim to detect are very subtle, it is essential to filter out all other regular or known variations in
this parameter. By analyzing each month and hour individually, we can eliminate the seasonal and diurnal components of foF2
variation. This approach assumes that the remaining variability is primarily due to solar and geomagnetic activity along with
random noise inherent in any real-time series. The effect of solar and geomagnetic activity on each of these data series was
filtered in the usual manner (e.g., Duran et al., 2023) by estimating the residuals ($foF2gs$) through a multiple regression
between foF2 and MgII (as a proxy for solar activity) and Ap, as follows:

106            $foF2gs = foF2exp - (A * MgII^2 + B * MgII + C * Ap + D)$           (1)

where *foF2exp* represents the measured foF2 data, and *A*, *B*, *C*, and *D* are the least squares parameters of the regression between
*foF2exp*, the linear and quadratic terms of MgII, and the Ap index.
Finally, the *foF2* linear trend, $\alpha$, is estimated from:

110            $foF2gs = \alpha t + \beta$                          (2)

where $\alpha$ in MHz/year and $\beta$ in MHz are the least squares parameters of the linear regression between foF2 and time *t* in year.

**4. Results and Discussion**
Figure 3 shows the values of the squared correlation coefficient, *r²*, between MgII, Ap, and foF2 indicating the fraction of foF2
monthly median variance explained by MgII index and Ap through equation (1) along the period 1960-2023, except for Port
Stanley, which covers the period 1960-2019.



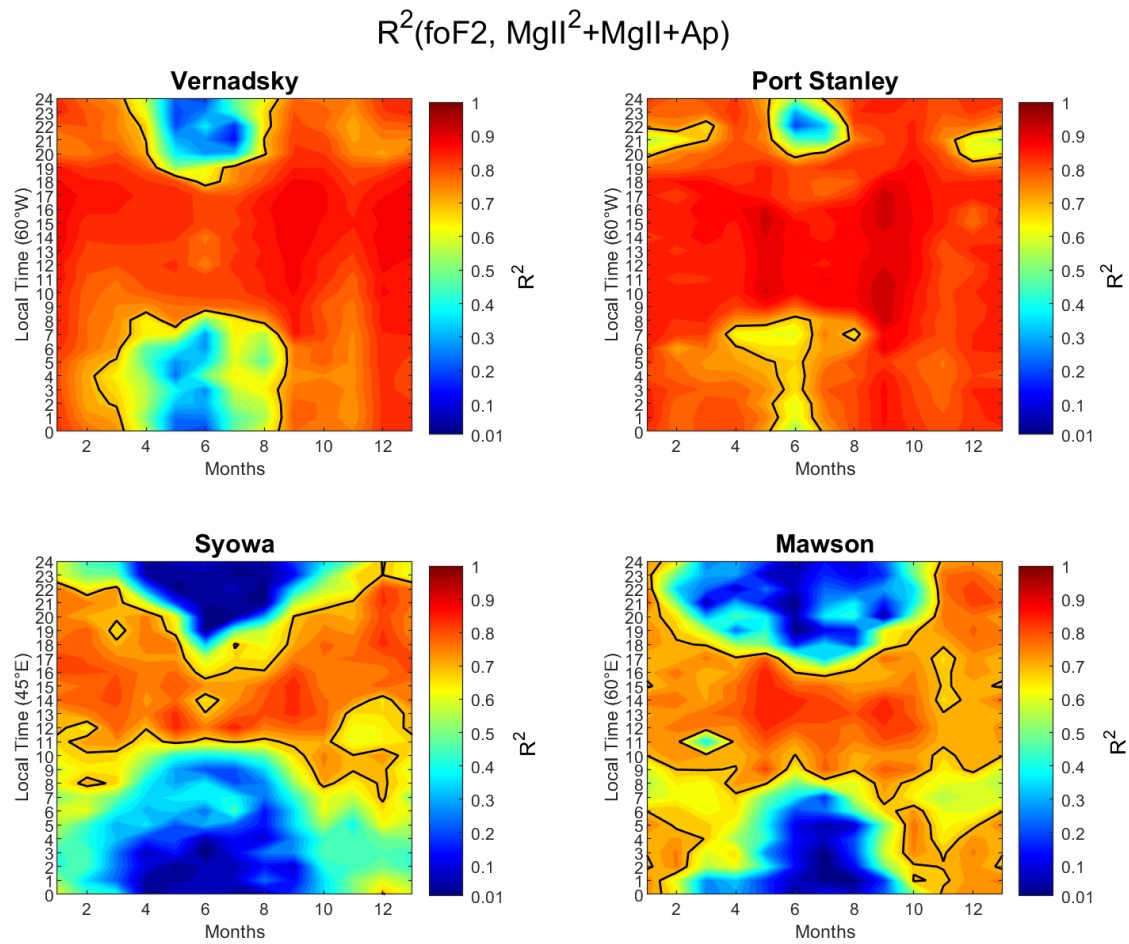

**Figure 3: Seasonal-diurnal variation of squared correlation coefficient ($r^2$) for: (a) Vernadsky, (b) Port Stanley, d) Syowa, and (c) Mawson. Solid black line: $r^2$=0.75**

Figure 3 illustrates a strong solar and geomagnetic dependence at Vernadsky during all hours in the summer, but only during daylight hours (08:00-19:00 LT) for the rest of the year. This is likely due to the presence of the WSA in the summer months. Port Stanley exhibits this strong dependence during almost all hours and across all seasons. This is due to its location at a lower latitude compared to the rest of the stations. A similar pattern to Vernadsky is observed at Mawson, although weaker correlations are observed during summer nights. In contrast, Syowa shows strong dependencies only during daylight hours and dusk, with almost null correlation during the period between 23:00 and 08:00 LT in winter months.

Trends are then calculated for all hours across all years included in the study following the equation (2). The results are displayed in Figure 4.



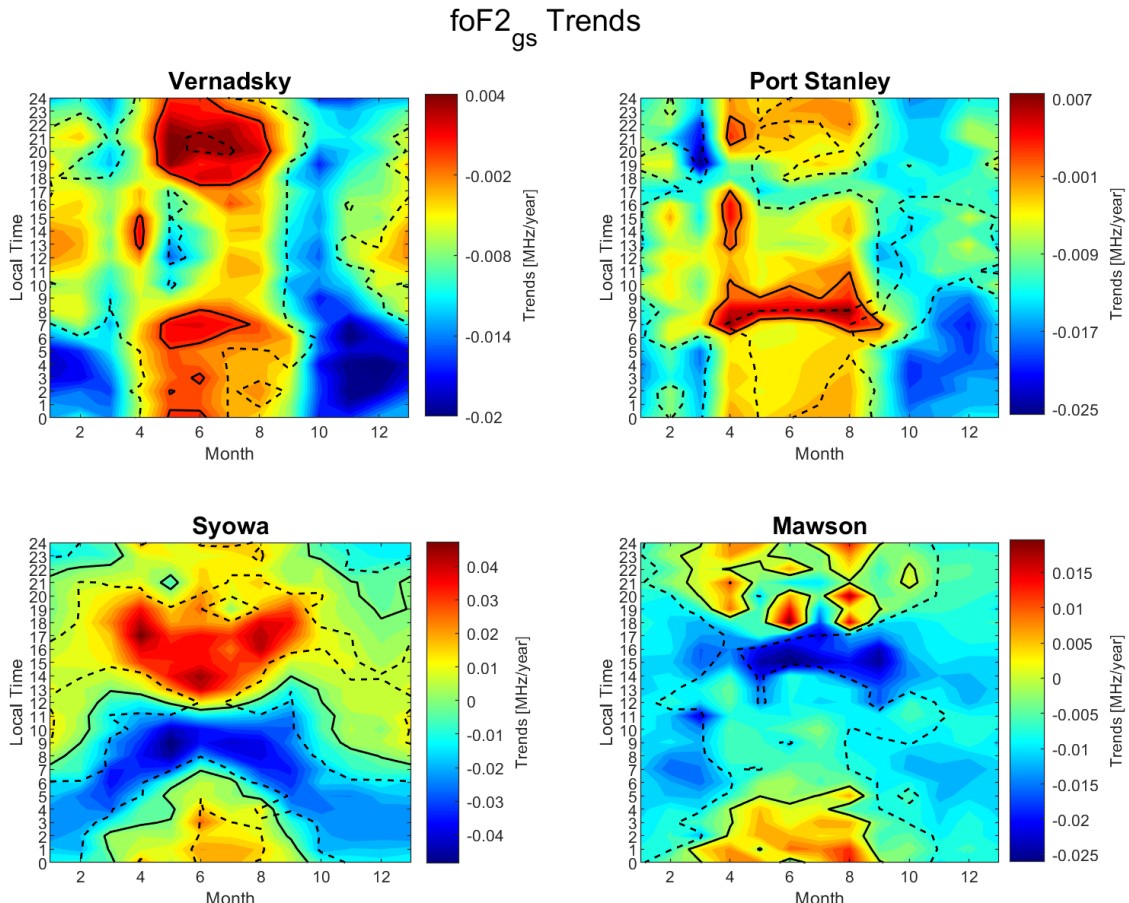

**Figure 4: Seasonal-diurnal variation of the foF2 linear trend for: (a) Vernadsky, (b) Port Stanley, c) Syowa, and (d) Mawson. Solid black line is $\alpha$=0. Dashed black line is $\alpha$ with 95% significance.**

Figure 4 shows negative trends (extreme values of -0.02 MHz/year) for Vernadsky in the intervals where the explained variance was significant, i.e., during all hours between end of September and December and at nighttime hours from January to March. Also a small positive trend during winter between 20:00 and 21:00 LT is observed. Similar negative trends are observed at Port Stanley at the same hours and months. Syowa and Mawson display opposite trends to each other. While Syowa shows positive trends (+0.04 MHz/year) during the intervals of significant explained variance, negative trends (-0.02 MHz/year) are observed for Mawson at the same hours.

The negative trends observed at the Vernadsky station coincide with the months and hours when the Weddell Sea anomaly is present. The physical phenomenon responsible for these trends during these intervals is likely the same as the one affecting Port Stanley. This can be inferred from the fact that Port Stanley exhibits a secondary maximum during the night (see Fig. 2).



Such a pattern is not observed at the Syowa and Mawson stations, which, despite their proximity (23° longitude apart), show
significantly different trends from each other.
Regarding the Weddell Sea anomaly and its impact on the ionosphere in this region, it is observed that it not only significantly
influences the Vernadsky station but also affects, albeit to a lesser extent, the Port Stanley station. As shown in Figure 2, during
the summer months, the foF2 parameter increases at night. The shape of the curve in this figure appears to combine the expected
trend in the absence of the anomaly (similar to the Mawson or Syowa curves) with a curve clearly influenced by the anomaly,
such as the Vernadsky curve. This behavior suggests that the Weddell Sea anomaly has a regionally differentiated effect,
modulating ionospheric characteristics based on the location of each station.
Moreover, since the WSA has been suggested to arise because the area is located farther away from the austral auroral zone
than locations at other longitude sectors along the same latitude (Richards et al., 2017 and 2018), the WSA may also depend
on the long term trend of the auroral zone itself. Indeed, perusal of the long term changes of the geomagnetic parameters in
the southern hemisphere using the International Geomagnetic Reference Field (IGRF20,
https://www.ngdc.noaa.gov/geomag/calculators/magcalc.shtml) shows that the austral auroral zone does move away from
middle latitudes at the American longitude sector between 1960 and 2024. So much so that the inclination of the magnetic
field at the WSA is almost stationary (in contrast to Concepción and Tucumán as seen in Foppiano et al., 1999) although the
total intensity of the field does decrease. This latter fact is related to the westward movement of the SAMA during the same
time interval.
In the case of Port Stanley, trends around -0.003 MHz/year are observed during the winter months between 10:00 and 14:00
LT, similar to the trends reported by Upadhyay and Mahajan (1998), who calculated a trend of -0.004 MHz/year for the period
from 1957 to 1990 during the same hours. However, between October and February, the trends calculated in this study are
lower than -0.005 MHz/year, reaching as low as -0.015 MHz/year in October (Fig. 4).
According to Danilov and Mikhailov (2001), using a third-degree polynomial on sunspot number to model foF2, the hourly
average trends for Vernadsky are negative throughout the day, which is consistent with the trends observed in this study, but
with different amplitudes (approximately half of the maximum value at 04:00 LT). When comparing Port Stanley, we again
find that the average hourly trends are negative for all hours, which is consistent with the trends in this study, but with double
the maximum values. On the other hand, when comparing the trends at 04:00 LT, neither Port Stanley nor Vernadsky show
differences in trends during the WSA months.
Syowa, according to model 1 of Alfonsi et al. (2001), shows negative trends in the monthly averages during the summer
months and positive trends during the winter, but with greater amplitude compared to this study. Model 1 consists in using
ITU-R global model to model foF2 and filter external forcings not linked to the greenhouse gas increase, which is a reason for
the difference in the trend absolute values between this study and Alfonsi et al. (2001). Specifically, in this study, between
23:00 and 07:00 LT, positive trends were found during the winter and negative trends during the summer. Then, between 08:00
and 13:00 LT, these trends reverse between winter and summer, and between 14:00 and 22:00 LT, they are positive throughout



the year. These same trends (model 1) for Mawson are negative throughout the year, consistent with this study, but with greater
amplitudes. During daylight hours, the trends are negative year-round, while at night, the trends in winter are close to zero.
The possibility of using time series only up to 2005 was evaluated, not only to maintain a single data source for Port Stanley,
but also because of solar minima that occurred after 2008, which could influence the trend results (Cnossen and Franzke.,
2014), obtaining similar results.
When comparing trends with those from other mid-latitude stations in South America, Foppiano et al. (1999) analyzed foF2
time series from the Concepción ionospheric station (36.8°S, 73°W) for the period 1958–1994. They found consistently
negative trends between 08:00 and 19:00 LT throughout the year. However, between 00:00 and 07:00 LT, the trends were
close to zero or positive, except during the summer months. Meanwhile, Jarvis et al. (1998), studying the trends (1957–1995)
in hmF2 at the Argentine Islands and Port Stanley, observed seasonal and diurnal variations. They reported predominantly
negative trends at Port Stanley, while smaller trends were noted at the Argentine Islands.
Several trend studies have been conducted on stations in the Southern Hemisphere. For example, Sharan & Kumar (2021) and
Duran et al. (2023) analyzed foF2 data at 00 and 12 LT from Australian ionospheric stations. In Sharan & Kumar (2021), foF2
data from Hobart, Canberra, and Christchurch (1947–2006) were examined. Their results revealed more significant trends at
midday (12 LT), with negative trends associated with F10.7 solar flux and small, insignificant positive trends linked to Rz.
They concluded that foF2 decreased by 0.1–0.4 MHz over five solar cycles, likely due to increased $CO_2$ in the troposphere
cooling the upper atmosphere. For its part, Duran et al. (2023) analyzed foF2 data from mid-to-low latitude stations up to 2022,
focusing on seasonal and diurnal variability. Their findings show overall negative trends, with the most significant declines
observed around the equinox. Weaker or slightly positive trends were seen in December–February and June–August, while
the diurnal pattern showed the strongest negative values during the day and the weakest at night.
To compare the experimental foF2 trend values with those from models assessing anthropogenic forcing effects, the results of
Solomon et al. (2018) are considered. They carried out simulations using the Whole Atmosphere Community Climate Model-
eXtended to investigate anthropogenic global changes across the entire atmosphere, including the thermosphere and
ionosphere, and identified $CO_2$ as the primary driver of temperature changes. For their simulations, they applied a $CO_2$
increase of 16 ppmv per decade, which led to a 1.2% reduction in peak electron density (NmF2). In this work, we find a foF2
maximum reduction of 3.5% per decade for Vernadsky and Port Stanley during months and hours of WSA. foF2 maximum
reductions of 10% and 6% were found for Syowa and Mawson, respectively, during other months and hours. All of these
percentages are much higher than those calculated in the literature (see De Haro Barbas & Elias, 2020; De Haro Barbas et al.,
2021; Duran et al., 2023).
The same long-term trend analysis has been performed but using F30 instead of MgII as an EUV solar proxy, as suggested by
recent studies (Laštovička and Burešová, 2023; Laštovička, 2024), however, the results (figure not shown) do not show
significant differences with those done with MgII.



**5. Conclusion**

The seasonal-diurnal variation of the long-term foF2 trend for stations within the Weddell Sea anomaly region is analyzed to contribute to the still controversial issue of the ionospheric trend. The WSA is shown to significantly impact ionospheric trends, particularly at Vernadsky, where negative trends are observed during periods when the WSA is active. This effect is also detected in Port Stanley, although to a lesser extent, showing only a secondary maximum during the evening. These trends seem to be consistent with the long-term apparent movement of the WSA relative to the austral auroral zone, which moves poleward during the studied time interval due to the decreasing of the total intensity of the magnetic field over the area.

The trends in foF2 show seasonal-diurnal variations, with negative trends at Vernadsky and Port Stanley during specific hours and months where the WSA is present. In contrast, Syowa and Mawson stations, in longitude sectors outside the WSA region, do not show the same seasonal-diurnal behavior of the trends.

The results are consistent with some earlier studies, though the observed trend magnitudes differ. For example, trends at Port Stanley match previous studies in terms of negative values, but with differing amplitudes. The study also aligns with findings from other Southern Hemisphere stations which report negative trends in foF2 at various latitudes.

Other studies suggest a 1.2% reduction in NmF2 due to $CO_2$-driven temperature changes. This study found foF2 maximum reductions values much larger than the literature at all stations. Particularly, Vernadsky and Port Stanley show the same maximum reductions values at WSA months. Overall, the study underscores the complex interplay between solar, geomagnetic, and regional factors in shaping ionospheric trends, with specific attention to the regional effects of the WSA.

**Author contribution**

MC: Formal analysis, data curation, writing – original draft preparation, validation. TD: Formal analysis, data curation, writing – original draft preparation. MB: Conceptualization, methodology, writing – original draft preparation. AZ: Data mining. AF: Supervision, writing – original draft preparation.

**Competing interests**

The authors declare that they have no conflict of interest.

**Acknowledgments**

We thank Ana G. Elias for her collaboration in the discussion and partial analysis of the results obtained. This work was supported by the Universidad Adventista de Chile, Regular Project number 204. The authors are thanks to the Antarctic



231   Geospace and ATmosphere reseArch (AGATA) Scientific Research Programme. M. Bravo acknowledges to

232   ANID/FONDECYT Regular 1211144.



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
