# Peer review of "Distinct ionospheric long-term trends in Antarctica due to the"

_EGUsphere, 2025_

## Referee Comment (RC1)

**Reviewer comments to the paper by Canales et al. "Distinct long-term trends…"**

The paper is dedicated to studying long-term trends in the critical frequency foF2 in the region of the Weddell Sea Anomaly (WSA) based on the vertical sounding observations at Argentine Island and Port Stanley stations located within that region.

The Introduction presents a brief description of the WAS event and previous attempts to study trends in the F2-layer parameters by various researchers. The hourly monthly medians of foF2 for 1960-2023/2019 are analyzed.

The authors illustrate the WAS effect in Fig. 2 comparing the diurnal variations in the foF2 medians for the analyzed stations and Syowa and Mawson stations located outside the WAS region.

The method of revealing long-term trends is the one used by many authors. The difference between the observed values of foF2 and its dependence on the MgII solar activity index and the Ap index was calculated. Linear regression between that difference and time was providing the sought for trend.

The comparison of charts of the correlation coefficient r between foF2 and combination of MgII and Ap in Fig. 3 demonstrates a visual difference between the stations within the WAS and outside it. The main difference is in the presence of high (>0.75) values of r at nighttime hours at the stations within WAS.

The main result of the study is shown in Fig. 4 in the form of charts with the foF2 trends as functions of the season and local time for all four stations. The charts demonstrate a complicated picture of regions with both positive and negative trends. The rest of the paper is dedicated to description of the features of the foF2 trend behavior in Fig. 4 at various locations and local times.

The principal conclusion is that WSA impacts significantly foF2 trends. For example, at Argentine Island, "… negative trends are observed during periods when the WSA is active…" The effect, although in weaker degree, is seen at Port Stanley.

It is found that "…the trends in foF2 show seasonal-diurnal variations …during specific hours and months where the WSA is present". No such variations are found for the two stations outside WSA.

I think that the paper presents an interesting step in consideration of a very important problem of ionospheric trends. I recommend the paper for publication with a minor revision.

My critical comments are as follows.

In Fig. 3 it is very difficult to read numbers at the ordinate. I recommend changing the step to 4 hours and make the letters larger.

Captions to Fig. 3: "Solid black line…". In my understanding of English, it is a curve, but not a line.

---

## Author Comment (AC2)

Dear Referee #2:

Their comments were pertinent to improving our manuscript, for which we are very grateful. Below are the responses to each of their observations in blue and the text that will be added to the manuscript is in red.

The paper is emphasizes important problem to studying long-term trends in the critical frequency foF2, which are the regional anomalies, specifically the region of the Weddell Sea Anomaly (WSA).

I have found the main conclusions well documented, but would suggest to reconsider some statements as indicated below:

1. 31 We detected two ionosondes -> We used two ionosondes
   Done.

2. 58 The extensive dataset of monthly median foF2 from the Vernadsky ionospheric station, covering the period from 1996 to 2023. -> add at the end "was built by including:"
   It has to be noted in the manuscript that most of the data were not manually scaled or evaluated, which can introduce significant errors even when using foF2 parameter which is well known even for modern digisondes operated at mid-latitudes. For ionosondes operated at polar cap or auroral ionospheric regions, the foF2 parameter is even less reliable.

   (it is not enough to be noted as per line 89: "Possible deficiencies and missing data from these stations are presumed to be due to their proximity to the auroral oval, a highly dynamic region where geomagnetic conditions can significantly interfere with ionospheric measurements." which is only explaining missing F-region data due to ionospheric absorption)

   New text has been added after the table to clarify that most of the data is manually scaled except for some time intervals at Port Stanley and Mawson stations. "It should be noted that most of the data are manually scaled. In particular, the year intervals used for Vernadsky and Syowa are entirely manually scaled. While for Mawson and Port Stanley, they are a combination of manual and autoscaling, the latter method has only been used for a little over

a decade. Furthermore, the significant errors introduced by this combination should be reduced by using monthly medians."

The figure below shows the consistency between the ionosonde and digisonde data for Port Stanley, even during a period where the two series intersect. The high variability of the auroral stations (Syowa and Mawson) can also be seen. Although Syowa is entirely manually scaled and Mawson only partially, both appear to behave similarly regardless of the scaling method.

[Figure]

3. random noise inherent in any real-time series -> random noise inherent in any physically measured series
   Done.

4. 120> Figure 3 illustrates a strong solar and geomagnetic dependence at Vernadsky during all hours in the summer, but only during daylight hours (08:00-19:00 LT) for the rest of the year. This is likely due to the presence of the WSA in the summer months
   I don't agree and have to suggest here that it could be simply due to polar-night effect (where there is long-term missing solar illumination of ionosphere around June at Vernadsky and other Antarctic stations, while Port Stanley is normal mid-lat station.)

   You are right. The sentences was modified including the polar night effect.

"The Antarctic stations (Vernadsky, Syowa, and Mawson) show low to almost zero correlations during winter nights due to the polar night effect. This also occurs during summer nights, with the exception of Vernadsky, which shows a high correlation during these months, probably due to the presence of the WSA."

5. Figure 4 - "Dashed black line is with 95% significance." As it is not very clear which areas are then significant and which not (which could be inferred only from the text), I'd suggest to make those not significant ones with lower hue/contrast (as those are not very relevant anyway).
Transparent white areas were added to Figure 4 and represent non-significant trends to make it easier for the reader.

6. 140:Such a pattern is not observed at the Syowa and Mawson stations, which, despite their proximity (140 23° longitude apart), show significantly different trends from each other
-> replace the part "despite their proximity (23° longitude apart)" by something like due to their differing prevailing placements within auroral and polar cap ionosphere, respectively,...
Done